# The GHGs Evolution of LULUCF Sector at the European Union (EU-27 + UK): Romania Case Study

Mihaela Iordache [1], Felicia Bucura [1], Roxana Elena Ionete [1,*], Remus Grigorescu [2], Andreea Maria Iordache [1], Ramona Zgavarogea [1], Alin Chitu [1], Anca Zaharioiu [1], Oana Romina Botoran [1,*] and Marius Constantinescu [1,*]

[1] ICSI Analytics Group, National Research and Development Institute for Cryogenics and Isotopic Technologies—ICSI, 4 Uzinei Street, 240050 Râmnicu Vâlcea, Romania

[2] Economic Sciences Department, Constantin Brancoveanu University, Bulevardul Nicolae Bălcescu 39, 240177 Râmnicu Vâlcea, Romania

* Correspondence: roxana.ionete@icsi.ro (R.E.I.); oana.dinca@icsi.ro (O.R.B.); marius.constantinescu@icsi.ro (M.C.)

**Abstract:** Mitigating climate change is a challenge that urgently needs to be addressed, as it has an increasing impact on the planet. According to the latest reports, global $CO_2$ emissions must be neutralized by 2050 in order to limit the rise in temperature to 1.5 °C. This work presents the evolution of Land Use, Land Use Change and Forestry (LULUCF) greenhouse gas (GHG) emissions/removals at the EU-27 + UK level for the 1990–2019 time period, as well as LULUCF emissions/removals forecasts for Romania up to 2040. The results revealed a 23% reduction in GHG emissions for the EU-27 + UK in 2019 compared to 1990. Romania's yearly average of GHG emissions/removals was 28,000 kt $CO_2$ eq., representing roughly 9.7% of the EU's annual average. In terms of projections for Romania, the only scenario that will not be in the target set by the new LULUCF Regulation is WEM (Reference Scenario/With Existing Measures), in which net GHG removals will be reduced by approximately 218 kt $CO_2$ eq., or 0.9 percent, in 2030 compared to the reference year; in 2040 compared to 1989, the trend will be accentuated both in absolute values, with a decrease of over 3000 kt $CO_2$ eq., and in relative values of 12%.

**Keywords:** GHG emissions/removals; LULUCF sector; climate change; mitigation

## 1. Introduction

For most researchers throughout the world, it is no longer a secret that climate change is caused by anthropogenic activities, and that high levels of GHG emissions are a consequence of them, which is a cause for concern [1]. Among the GHGs, $CO_2$ is the main cause of the emergence and acceleration of global climate change with all the devastating consequences for humanity (e.g., rising global temperatures, rising oceans and sea levels, floods, wildfires, droughts, etc.) [2]. Currently, the $CO_2$ level has exceeded 415 ppm, and in the absence of major GHG mitigation actions, the global temperature is about to increase by an average of 6 °C (10.8 °F), according to the latest estimates. Climate change and environmental degradation are existential threats to the EU as well as to the whole world [3,4]. The 1992 Rio Protocol was the first to provide an overview of the global behavior of the world's states [5]. The Kyoto Protocol has continued this trend towards accountability, with numerically defined targets and specific mechanisms acting between governments. In 2015, world leaders agreed on ambitious new goals for the fight against climate change [6]. The Paris Agreement came into force on 4 November 2016, after meeting the conditions of ratification by 55 countries responsible for at least 55% of GHG emissions and providing an action plan to limit global warming [7,8]. With the adoption of the Paris Agreement, countries have committed to retaining the average global warming increase well below 2 °C above pre-industrial levels and to pursuing efforts to restrict it to 1.5 °C to prevent dangerous impacts of climate change [9,10].

To overcome these challenges, the EU needs a new growth strategy to transform it into a modern, resource-efficient and competitive economy in which net GHG emissions will be "zero" by 2050. The European Green Deal is the EU's plan for a sustainable economy. This requires that current GHG levels fall significantly over the next several decades. As an intermediate step toward climate neutrality, the EU has raised its ambition in terms of climate by 2030, committing itself to cutting emissions by at least 55% by 2030, under the so-called 'Fit for 55' packages, in order to align the current regulations with targets set for 2030 and 2050 [11,12]. Limiting warming to 1.5 °C requires a rapid, far-reaching and unprecedented transition in energy, land use, urban, infrastructure (including transport and buildings) and industry systems to substantially reduce emissions, based on an increase in investments in a broad portfolio of mitigation options [13–18]. As the role of the LULUCF sector in mitigating climate change for the next decades will be extremely important, a review of the LULUCF Regulation GHG emissions/removals is underway for the 'Fit for 55' packages. The main objective of the Commission's proposal is to strengthen the contribution of the LULUCF sector to the EU's overall increased climate ambitions for 2030 by setting an EU-level target of 310,000 kt $CO_2$ eq. in net GHG removals in the LULUCF sector in 2030 by the Emissions Trading System (ETS) and the Effort Sharing Regulation (ESR). The right rules will ensure that LULUCF fully contributes to the emissions reduction targets, while supporting the EU's biodiversity goals. In addition, from 2031, the EU wants to merge agriculture sector non-$CO_2$ emissions with the GHG emissions/removals under the LULUCF regulation to create a new single pillar covering AFOLU (Agriculture, Forestry and Other Land Use), with the goal of achieving climate neutrality by 2035 at the EU level in the combined sector [19]. Studies have shown that globally, agriculture contributes up to 12% of GHG emissions. Of these, 7.7% were issued by EU Member States. In the EU, 10% of total GHG emissions come from agricultural sources, with significant variation between Member States [20]. Forests and Harvested Wood Products (HWP) can also contribute positively to the achievement of the long-term goals set out in the Paris Agreement. The mitigation impact of the forestry sector includes direct carbon emissions from forests averaging around 400,000 kt $CO_2$/year from 2000 to 2015 (i.e., about 10% of total GHG emissions from the EU), minimizing the potential for HWP (Harvested Wood Products)—approximately 45,000 kt $CO_2$ over the same period and the potential substitution benefits of using wood products instead of GHG-intensive raw materials and energy sources [21].

The socio-economic and demographic changes taking place around the world have also combined with global technological changes, driven in part by policies that mitigate climate change and air pollution, in addition to market conditions. Romania, through its Recovery and Resilience Facility, seeks to mitigate the economic and social impacts of climate change and to create a more sustainable, resilient and better-prepared economy and society for the challenges and opportunities of the green and digital transitions [22].

Starting from this context, the purpose of this work is to show the evolution, for the 2020–2040 time period, of the GHG emission/removal projections in the LULUCF sector of the EU-27 + UK, especially Romania.

## 2. Methodology

For the analysis of the LULUCF sector, the reference year for Romania is 1989, and 1990 for the EU-27 + UK; 2005 is the year the EU began to consistently implement a number of GHG removals policies and measures to meet its targets. Since 2012, data/information are used to generate activity data (kha) by processing maps through complex query and intersection processes, respectively using a type 3 approach/explicit geospatial, LPIS/IACS + CLC (Corine Land Cover) + LiDAR and aero-photogrammetry technologies [23]. In this respect, the analysis considered three time periods: (i) 1989–2005 (Romania) and 1990–2005 (EU-27 + UK); (ii) 2005–2012; (iii) 2012–2019. The GHG emissions/removals are expressed as a simple arithmetic mean, and the specific dynamics either for the defined time periods or for the analyzed time period are highlighted with the help

of the average indicators of the time series: (i) the absolute average change and (ii) the rate of the relative average change.

Over time, research on the GHG emissions/removals evolution/projection has considered the following: (i) synthetic characterization of the evolution through relevant indicators; (ii) detachment of systematic elements that reveal patterns or repeatability of evolution over time; (iii) study and interpretation of the past evolution depending on the dynamics of land use categories.

The projections were established based on the National Greenhouse Gases Inventory (NGHGI) historical data/information [23]. For the projection of the land use area, we applied: (i) the absolute mean change method if the projection was increasing; (ii) the mean index method if the projection was decreasing. Theoretically, the determination of the projection through the method of absolute mean change implies that the adjusted values are determined with the following recurrence relation:

$$\hat{y}_k = y_0 + k \cdot \overline{\Delta}, k = \overline{0, T} \tag{1}$$

where $y_0$ is the term considered as the basis of adjustment (year 1989), $k$ is the temporal series, $T$ is the unit of time and $\overline{\Delta}$ represents the absolute mean change which is calculated as the simple arithmetic mean of the absolute changes based on the chain:

$$\overline{\Delta} = \frac{(y_1 - y_0) + (y_2 - y_1) + \ldots + (y_{T-1} - y_{T-2}) + (y_T - y_{T-1})}{T - 1} = \frac{y_T - y_0}{T - 1} \tag{2}$$

The first and last adjusted values are equal to the first and last empirical values as follows:

$$\text{for } k = 0 \rightarrow \hat{y}_0 = y_0 + 0 \cdot \overline{\Delta} = y_0 \tag{3}$$

$$\text{for } k = T \rightarrow \hat{y}_T = y_1 + (T - 1) \cdot \overline{\Delta} = y_1 + (T - 1) \cdot \frac{y_T - y_0}{T - 1} = y_T \tag{4}$$

For extrapolation, the $k$ series is continued until the last unit of time related to the researched horizon—the year 2040. The evolution of the surfaces related to the specified areas has a linear projection over time that will be estimated by the method of absolute mean change.

The average index method is applied if the surfaces have a decreasing trend and the projection by the average increase method generates negative terms or equal to zero. The adjusted values are determined with the following recurrence ratio:

$$\hat{y}_k = y_1 \cdot \overline{I}^{k-1}, k = \overline{1, T} \tag{5}$$

where $y_1$ is the term considered as the basis for adjustment (year 1989 in the reference scenario/WEM, year 2005 in the case scenario with measures/WAM and year 2012 in the case of the scenario with additional measures/WOM) and represents the average index calculated as the simple geometric mean of relative changes based on in the chain:

$$\overline{I} = \sqrt[T-1]{\frac{y_2}{y_1} \cdot \frac{y_3}{y_2} \cdot \frac{y_4}{y_5} \ldots \ldots \frac{y_{T-1}}{y_{T-2}} \cdot \frac{y_T}{y_{T-1}}} = \sqrt[T-1]{\frac{y_T}{y_1}} \tag{6}$$

The first and last adjusted values are equal to the first and last empirical values as follows:

$$\text{for } k = 1 \rightarrow \hat{y}_1 = y_1 \overline{I}^{1-1} = y_1; \tag{7}$$

$$\text{for } k = T \rightarrow \hat{y}_T = y_1 \cdot \overline{I}^{T-1} = y_1 \cdot \left( \sqrt[T-1]{\frac{y_T}{y_1}} \right)^{T-1} = y_T \tag{8}$$

Based on the index, the average rate ($R$) with the relation is calculated:

$$\overline{R}(\%) = \overline{I}(\%) - 100$$

and together with the average level and the absolute average change, it contributes to the statistical characterization of the time series. Once the surfaces have been estimated, we start from the hypothesis that the quantitative level of GHG depends on the surface of the specified area and/or on the time variable, a function that is determined using the regression method. The regression method is a statistical approach for determining the connection between variables using mathematical functions, called regression functions. A mathematical expression obtained from the processing of experimental data that approximates the interdependence between two or more variables of a system or process is referred to as a regression function. When the connections between the various variables cannot be demonstrated theoretically accurately enough, a regression function must be determined. The following processes were involved in estimating a mathematical model that was linked to a certain type of projection: (i) Intuition of mathematical relations of the models based on the graphical representation of the correlation; (ii) Application of the Pearson coefficient to determine the intensity and direction of the link between GHG emissions/removals as a function of time and/or specific areas of different land categories. The Pearson coefficient ranges from $-1$ to $+1$; the closer it is to zero, the lower the bond's intensity, and the closer it is to one, the stronger the relationship. The relationship is directly proportional if the values are greater than zero, and inversely proportional if the values are negative; (iii) Estimating the parameters for each model, most often using the least squares method; (iv) Choosing the model that most accurately represents the relationship highlighted by the data (minimum criterion); (v) Testing the significance of the chosen model and the coefficients of the functions found; (vi) Assessing the model's significance and the coefficients of the functions identified; (vii) Economic interpretation of the tested parameters; and (viii) Use of the model for simulations and projections. There is one-factor regression (single) and multi-factor regression (multiple), depending on the number of influencing factors for the resulting characteristics. Unifactorial regression considers quantifying dependencies in the form of unifactorial models, testing hypotheses, and demonstrating predictive calculations. The unifactorial model describes the relationship between the two variables, $x$ and $y$, the other factors being considered with constant action. The theoretical equation of this regression is of the form:

$$y_i = f(x_i) + u_i, i = \overline{1, n} \tag{9}$$

where $u_i$ it represents the action of other factors (disturbance), and $n$ the number of observations. The estimation of the parameters of this model is done using the least squares method, which involves minimizing the sum of the squares of the deviations of the empirical values ($y_i$) from the estimated values ($\hat{y}_i$):

$$\sum_{i=1}^{n} (y_i - \hat{y}_i)^2 \to 0 \tag{10}$$

In unifactorial linear models, the resultant variable depends on a single factor by the relation:

$$y_i = a + b \times x_i + u_i, i = \overline{1, n} \tag{11}$$

The estimation of the parameters of this model can also be done using the least squares method, which involves minimizing the sum of the squares deviations of the empirical values ($y_i$) from the estimated values ($\hat{y}_i$):

$$\sum_{i=1}^{n} (y_i - \hat{y}_i)^2 = \sum_{i=1}^{n} \left(y_i - \hat{a} - \hat{b} \times x_i\right)^2 \to 0 \tag{12}$$

where $\hat{a}$ and $\hat{b}$ are the estimators of the linear model parameters. Minimizing the amount involves determining the stationary points that result from solving the system obtained by canceling the partial derivatives of the function:

$$F\left(\hat{a},\hat{b}\right) = \sum_{i=1}^{n} \left(y_i - \hat{a} - \hat{b} \times x_i\right)^2 \tag{13}$$

we arrive at the following system of equations:

$$\begin{cases} n\hat{a} + \hat{b}\sum x_i = \sum y_i \\ \hat{a}\sum x_i + \hat{b}\sum x_i^2 = \sum x_i y_i \end{cases} \tag{14}$$

and after performing the calculations the following estimation relations are obtained:

$$\hat{a} = \frac{\sum_{i=1}^{n} y_i \sum_{i=1}^{n} x_i^2 - \sum_{i=1}^{n} x_i \sum_{i=1}^{n} x_i y_i}{n\sum_{i=1}^{n} x_i^2 - \left(\sum_{i=1}^{n} x_i\right)^2} \tag{15}$$

$$\hat{b} = \frac{n\sum_{i=1}^{n} x_i y_i - \sum_{i=1}^{n} x_i \sum_{i=1}^{n} y_i}{n\sum_{i=1}^{n} x_i^2 - \left(\sum_{i=1}^{n} x_i\right)^2} \tag{16}$$

In the case of multifactorial regression, a resultant variable is defined according to several factorial variables which imply the use of multifactorial regression:

$$y = f(x_1, x_2, ..., x_n) + u_i, i = \overline{1, n} \tag{17}$$

where $u_i$ it represents the action of other factors (disturbance), and $n$ the number of observations. If the connection between each factor and the resultant variable is linear, then the regression function becomes:

$$y_{x_1, x_2, ..., x_n} = a_0 + a_1 \times x_1 + a_2 \times x_2 + ... + a_n \times x_n \tag{18}$$

in which $a_0$ is the parameter that concentrates the influence of the unregistered factors, considered with constant action; $x_1, x_2, ..., x_n$ are the factorial variables included in the research report; and $a_1, a_2, ..., a_n$ are regression coefficients, which show how much the resultant variable changes when the factorial variable changes by one unit.

The estimation of the parameters of this model can be done using the least squares method, which involves minimizing the sum of the squares of the deviations of the empirical values ($y_i$) from the estimated values ($\hat{y}_i$):

$$\sum_i (y_i - \hat{y}_i)^2 = \sum_i (y_i - \hat{a}_0 - \hat{a}_1 \times x_1 - \hat{a}_2 \times x_2 - ... - \hat{a}_n \times x_n)^2 \tag{19}$$

where $(\hat{a}_i)_{i=\overline{0,n}}$ are the estimators of the linear model parameters.

Minimizing the amount involves determining the stationary points that result from solving the system obtained by canceling the partial derivatives of the function:

$$F(\hat{a}_0, \hat{a}_1, \hat{a}_2, ..., \hat{a}_n) \sum_i (y_i - \hat{a}_0 - \hat{a}_1 \times x_1 - \hat{a}_2 \times x_2 - ... - \hat{a}_n \times x_n)^2 \to 0 \tag{20}$$

The following system of equations is constructed based on empirical data:

$$\begin{cases} n \times a_0 + a_1 \times \sum_i x_{i1} + a_2 \times \sum_i x_{i2} + ... + a_n \times \sum_i x_{in} = \sum_i y_i \\ a_0 \times \sum_i x_{i1} + a_1 \times \sum_i x_{i1}^2 + a_2 \times \sum_i x_{i1} \times x_{i2} + ... + a_n \times \sum_i x_{i1} \times x_{in} = \sum_i x_{i1} \times y_i \\ \cdots\cdots\cdots\cdots\cdots\cdots\cdots\cdots\cdots\cdots\cdots\cdots\cdots\cdots\cdots\cdots \\ a_0 \times \sum_i x_{in} + a_1 \times \sum_i x_{i1}^2 + a_2 \times \sum_i x_{in} \times x_{i2} + ... + a_n \times \sum_i x_{in}^2 = \sum_i x_{in} \times y_i \end{cases} \tag{21}$$

Based on the theoretical elements presented, it was necessary to decide on the statistical significance of the regression parameters and the validity of the adopted model. If the regression coefficients of the estimated models belong to a confidence interval that also contains the value of 0, the model was abandoned. After verifying the significance of the parameters, the next step was to validate the statistically specified model using the ANOVA test. Validating the model, which was assumed to be significant for the regression parameters, was carried out by interpreting a specific *p*-value test. Once validated, the model was used in projection simulation.

## 3. Results and Discussion

### 3.1. LULUCF Sector GHG Emissions vs. GHG Removals Evolution (EU-27 + UK's)

The LULUCF sector net removals (EU-27 + UK) increased from around 193,000 kt $CO_2$ eq. to 328,900 kt $CO_2$ eq. in 1999, reaching over 243,000 kt $CO_2$ eq. in 2019 (Figure S1, Supplementary Materials). During the period 1990–2019, the average removal was about 289,769 kt $CO_2$ eq., increasing from one year to another by 30,400 kt $CO_2$ eq. From a relative perspective, removals increased 0.8%/year over the period 1990–2019. Compared to other NGHGI sectors, LULUCF can act as both a source and a sink. Table 1 temporarily shows the evolution of the ratio between GHG removals and GHG emissions specific to the LULUCF sector.

**Table 1.** GHG emissions vs. GHG removals evolution, by GHG type, EU-27 + UK [24].

| Year | | 1990 | 2005 | 2012 | 2019 |
|---|---|---|---|---|---|
| GHG emissions, kt $CO_2$ eq. | total | 197,194 | 170,615 | 162,473 | 151,741 |
| | $CO_2$ | 168,044 | 142,499 | 134,162 | 123,247 |
| | $CH_4$ | 12,857 | 12,224 | 12,467 | 114,10 |
| | $N_2O$ | 16,294 | 15,892 | 15,845 | 17,085 |
| $CO_2$ removals, kt $CO_2$ eq. | | −390,389 | −473,151 | −476,342 | −394,862 |
| GHG emissions vs. GHG removals ratio | | 1.98 | 2.77 | 2.93 | 2.60 |
| No. of GHG source type countries | | 8 | 6 | 5 | 6 |
| No. of GHG sink type countries | | 20 | 22 | 23 | 22 |

Note 1: data sources: https://ec.europa.eu/eurostat/data/database, accessed on 17 January 2022.

The increase in GHG removals is due to the faster decline in GHG emissions, approximately 0.9%/year, compared to the rate of increase in $CO_2$ emissions, approximately 0.04%/year, Figure 1. For the equivalence of GHG emissions vs. GHG removals generated by specific activities, Romania used GWP AR4 [25]. In 1990, the number of the EU-27 + UK countries that were net carbon sinks was 20 out of 28, and in 2019 their number increased to 22. Although the number of net carbon sink countries has not increased significantly in the last three decades, we can estimate that GHG emissions have fallen from around 197,000 kt $CO_2$ eq. in 1990 to about 152,000 kt $CO_2$ eq. in 2019.

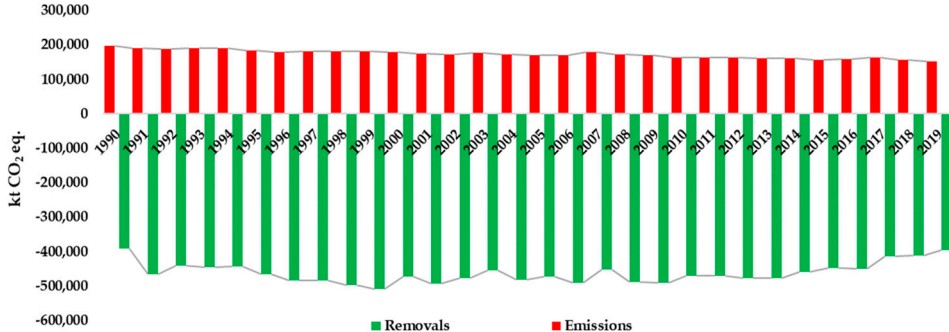

**Figure 1.** GHG emissions vs. GHG removals comparative dynamics, EU-27 + UK.

Basically, the GHG emissions decreased by 23% in 2019 compared to 1990. From 390,000 kt $CO_2$ eq. in 1990, the GHG removals had a smooth increase, to 394,000 kt $CO_2$ eq. in 2019, about 1.1% over time.

The share of $CO_2$ emissions in total LULUCF sector GHG emissions decreased from 85.2% in 1990 to 81.2% in 2019 (Figure S2, Supplementary Materials). In 2019, compared to 1990, the volume of emissions decreased by 26.7%. The share of $CH_4$ in total LULUCF GHG emissions increased from 6.5% in 1990 to 7.7% in 2012, after which it decreased slightly to 7.5% in 2019. The volume of $CH_4$ emissions remained relatively constant for 1990–2005. After 2005, $CH_4$ emissions fell to 11,410 kt $CO_2$ eq. and in 2019, compared to 1990, the volume of $CH_4$ emissions decreased by 11.3%. The share of $N_2O$ has steadily increased from 9.7% in 1990 to about 14% in 2019. The volume of $N_2O$ emissions has increased from 16,300 kt $CO_2$ eq. in 1990 to 17000 kt $CO_2$ eq., a growth trend that in relative terms increased in 2019 compared to 1990 by about 5%.

The trends highlighted above resulted in a decrease of 23% in 2019 compared to 1990 of GHG emissions, EU-27 + UK (Figure 2).

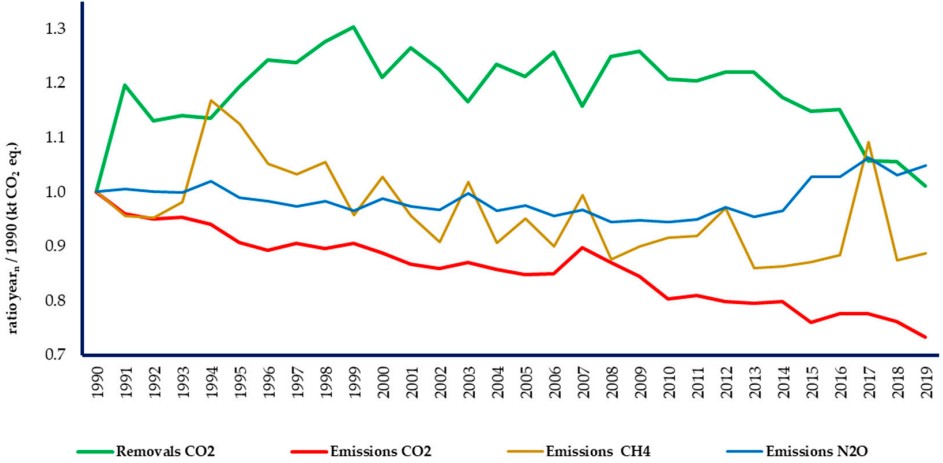

**Figure 2.** GHG emissions vs. GHG removals dynamics, by GHG type, EU-27 + UK. 1990 = 1.

The observed decrease in GHG removals, with an annual average rate of 2.64% per year in the 2012–2019 time period, attenuated the increased trends manifested during the intervals 1990–2005 (1.29%/year) and 2005–2012 (0.1%/year), with an average annual increased rate of 0.04% for the three decades 1990–2019 (Figure S4, Supplementary Materials). The decreasing trends of GHG emissions recorded during the three time periods highlighted determined a decrease of 1.06%/year for the whole analyzed time period. Forest land and HWP make the largest contribution to the GHG net removals budget, Table 2.

**Table 2.** GHG emissions/removals evolution, by type of land use categories, EU-27 + UK (kt $CO_2$ eq.).

| Year | 4A. FL | 4B. CL | 4C. GL | 4D. WL | 4E. SL | 4F. OL | 4G. HWP | 4H. Other |
|------|--------|--------|--------|--------|--------|--------|---------|-----------|
| 1990 | −347,716 | 87,835 | 36,278 | 15,594 | 41,919 | 3892 | −31,260 | 0.263 |
| 2005 | −403,043 | 70,414 | 19,063 | 19,877 | 48,020 | 1243 | −58,989 | 0.879 |
| 2012 | −429,771 | 63,229 | 20,482 | 17,282 | 48,897 | 1190 | −35,780 | 0.603 |
| 2019 | −344,619 | 57,038 | 11,884 | 20,421 | 50,313 | 1728 | −40,412 | 0.526 |

Note 2: 4–LULUCF sector; 4A. FL—Forestlands; 4B. CL—Croplands; 4C. GL—Grasslands; 4D. WL—Wetlands; 4E. SL—Settlements; 4F. OL—Other Lands; 4G. HWP—Harvested Wood Products; 4H. Other.

GHG emissions from Cropland have been steadily declining at an average annual rate of 1.5%. From Grassland-specific activities, the GHG emissions declined at an average annual rate of 4.2% until 2005 and tended to rise slightly at an average annual rate of 1%

between 2005 and 2012. After 2012, emissions from Grassland fell sharply from year to year at an average annual rate of 7.5%. Wetlands recorded increasing trends in GHG emissions from 1990 to 2005 at an average annual rate of 1.6% and from 2012 to 2019 at an average annual rate of 2.4%. In the period 2005–2012, emissions from Wetlands decreased at an average rate of 2%. The effect of these developments determined an increasing trend for the analyzed time horizon by 0.9%/year. According to Figure 3, Settlements generated GHG emissions that increased from one year to another, from 1990 to 2019, at an average annual rate of 0.6%/year. A summary of the GHG emissions from the LULUCF sector is given in Table S1 (Supplementary Materials).

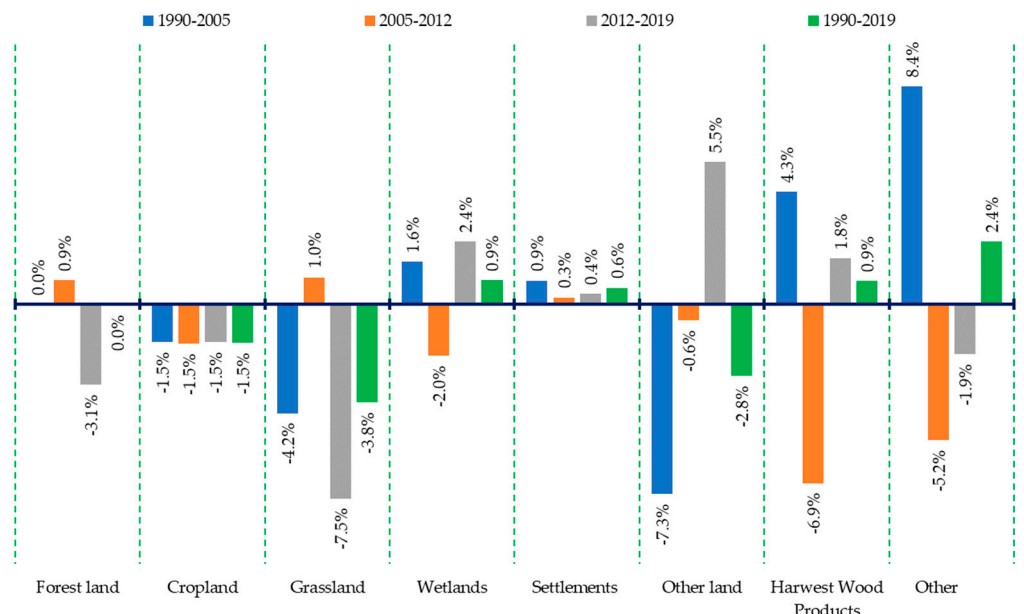

**Figure 3.** The GHG emissions/removals dynamics, by land use, EU-27 + UK.

　　　GHG emissions from the Settlements increased by 20% in 2019 compared to 1990. GHG emissions generated by Wetlands of the analyzed EU area increased by 31% in 2019 compared to 1990, and GHG emissions from Other doubled in 2019 compared to 1990. GHG emissions with increasing trends over the last 30 years had an average share of total GHG emissions in the LULUCF sector of about 38%. The three categories generated an increase in GHG emissions by approximately 13,500 kt $CO_2$ eq. in 2019 compared to 1990. The year-on-year average annual growth rate of GHG emissions generated by the three categories was 0.7%/year, an average annual increase, in absolute numbers, of 465 kt $CO_2$ eq./year. GHG emissions generated by Cropland, Grassland, Forestland and Other land use categories showed decreasing trends over the analyzed time horizon. Thus, GHG emissions decreased across all listed land categories by approximately 59,000 kt $CO_2$ eq. in 2019 compared to 1990. This decreasing trend resulted in a year-on-year decrease of 1.9%/year, a decrease of just over 2000 kt $CO_2$ eq. The share over the 30 years of emissions generated by land use categories with increasing trends in total emissions was 62%.

　　　In Romania, agricultural activities, mainly from land management and land use change in cultivated land, are identified as one of the main producers of greenhouse gases (GHGs) from LULUCF, namely methane ($CH_4$), nitrous oxide ($N_2O$) and carbon dioxide ($CO_2$), accounting for more than 40% of GHG emissions (Figure 4). That is why the intelligent management practices of cultivated lands, among which we mention the replacement of synthetic nitrogen with organic fertilizer, the use of crops with a high capacity to fix nitrogen in the soil, and encouraging organic farming, could be measures to reduce GHG emissions.

　　　The overall decreasing trend of GHG emissions generated by LULUCF over the analyzed period was determined by the average annual rate of decrease of −1.9% of GHG emissions generated by Cropland, Grassland, Forestland and Other land, in absolute value,

then the average growth rate of + 0.7% of emissions was generated by the following land categories: Settlements, Wetlands and Other land. From year to year, the decreasing trends of the listed categories expressed in absolute values was just over 2000 kt $CO_2$ eq./year, a value that is over 4.5 times higher than the average annual increase of 465 kt $CO_2$ eq. recorded cumulatively for land with an upward trend over the analyzed time horizon. $CO_2$ removals were generated by the following land use categories: Forestland and HWP. In summary, the evolution of GHG removals is presented in Table 3.

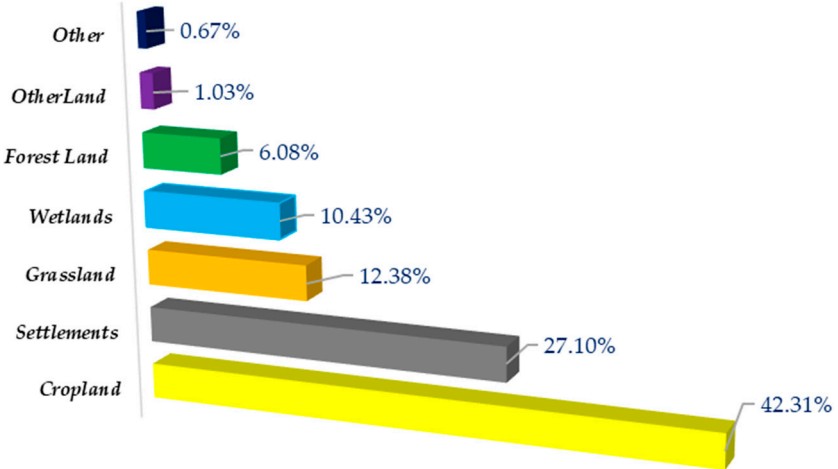

**Figure 4.** LULUCF GHG emissions structure by land use categories for the 1990–2019 time period, EU-27 + UK.

**Table 3.** LULUCF GHG removals structure by land use categories during 1990–2019, EU-27 + UK.

| | Year | | | | $CO_2$ Removals | | |
|---|---|---|---|---|---|---|---|
| | **1990 (kt $CO_2$ eq.)** | **2005 (kt $CO_2$ eq.)** | **2012 (kt $CO_2$ eq.)** | **2019 (kt $CO_2$ eq.)** | **Average Level (kt $CO_2$ eq.)** | **Rhythm of Change— 2019/1990 (%)** | **GHG Removals Structure (%)** |
| Forestland | −359,129 | −413,974 | −440,411 | −354,450 | −422,847 | 1.3 | 91.42 |
| Harvested Wood Products | −31,260 | −58,989 | −35,780 | −40,412 | −39,618 | 29.3 | 8.57 |
| Other land | 0 | −0.188 | −0.150 | 0 | −0.086 | - | 0.02 |

In comparison, the GHG removal levels showed relative uptrends in 1990–1999 and 2007–2009, interspersed with sub-periods having slightly downtrends (2001–2003 and 2013–2019). For the LULUCF sector, the most important contribution to the negative emissions budget had the land categories: Forest land remaining Forest land, land in conversion to Forest land and Cropland remaining Cropland. In contrast, the largest sources of GHG emissions are the conversion of land into Other land and land conversion into Settlements.

*3.2. Romanian LULUCF Sector GHG Emissions vs. GHG Removals Evolution*

The Romanian's LULUCF sector GHG emissions/removals have increased since 1989 from ~25,200 kt $CO_2$ eq. to ~30,200 kt $CO_2$ eq. in 2019, an increase of approximately 20% which represents an increase of 5000 kt $CO_2$ eq. in 2019 compared to the reference year. The annual average of GHG emissions/removals was approximately 28,000 kt $CO_2$ eq., representing approximately 9.7% of the annual average of GHGs at the EU countries level, approximately 290,000 kt $CO_2$ eq. calculated for the 1990–2019 time period. Romania, along with 22 other countries, is a GHG net sink, Figure 5.

Net GHG removals had tendencies to increase at EU-27 + UK and Romania levels; Romania tended to have fewer fluctuations compared to the EU region. For EU-27 + UK,

the average annual growth rate of 0.8%/year during the 1989–2019 time period is higher than the average annual growth rate recorded by Romania, at +0.6%/year (Figure 6).

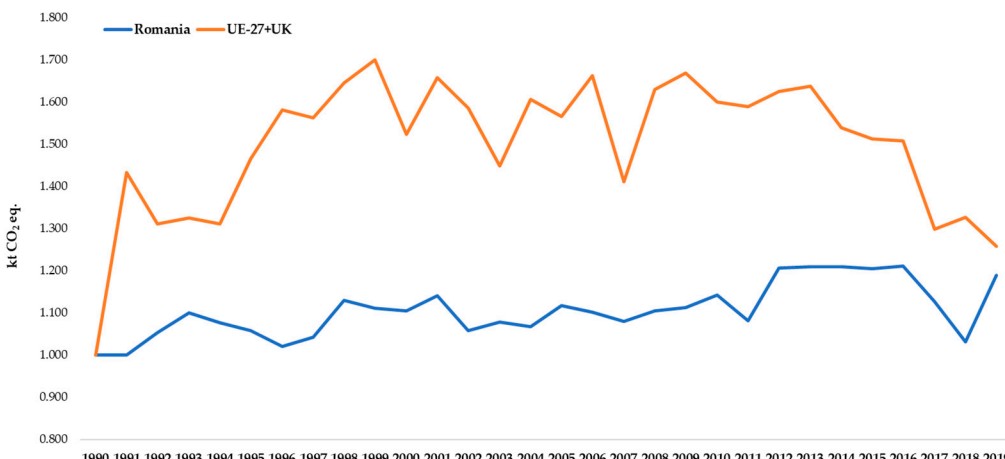

**Figure 5.** Net GHG removals dynamics comparative aspects, UK-27 + UK vs. Romania (1990 = 1).

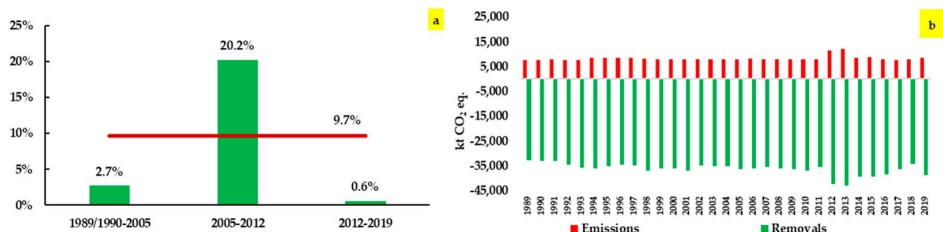

**Figure 6.** (**a**) Contribution of the Romanian LULUCF sector to net GHG removals from EU-27 + United Kingdom (Calculation was made as a percentage ratio between the absolute change in removals in Romania and the absolute change in net removals at the UK-27 + UK's level); (**b**) GHG emissions vs. GHG removals comparative dynamics for Romania.

Net GHG removals average in Romania increased in the 1989–2005 period by 198 thousand tons CO$_2$ eq., i.e., by 0.7%. In comparison, in the EU-27 + UK, net GHG removals increased at an average rate of 3%/year, higher than that recorded by Romania. The consequence of this situation determined a contribution of Romania to the absolute change of the net GHG removals in the European countries by 2.7%, well below the sectoral contribution of Romania registered for the investigated time period. Between 2005 and 2012, the net GHG removals increased by 1.1%/year in Romania, a higher rate than that recorded in the EU, +0.5%/year. This accentuated upward trend had the effect of doubling Romania's contribution to the absolute change in net GHG removals compared to that recorded in the period 1989–2019, 20.2%, Figure 6.

Between 2012 and 2019, net GHG removal decreased in both Romania and EU27 + UK. In Romania, the rate of decrease was 0.2%/year, lower than the 3.6%/year in EU27 + UK. During this period, Romania recorded the smallest contribution to the absolute change in net GHG emissions at EU27 + UK levels. A summary analysis of the Romanian LULUCF sector for the 1989-2019 time period on time segments influenced by the policies and measures from the state authorities' level is shown in Table 4.

The GHG removals average between 1989 and 2019 was about 36,300 kt CO$_2$ eq., exceeding about 4.4 times the GHG emissions average. GHG removals have increased by more than 200 kt CO$_2$ eq., higher than 5.9 times the annual absolute average change in GHG emissions. GHG removals have increased by an average of 0.6%/year, more than the growth rate of GHG emissions. The analysis by time segments shows that GHG emissions increased faster, +5.4%/year, than GHG removals, +2.2%/year, between 2005 and 2012. However, GHG removals have increased year-to-year by 839 kt CO$_2$ eq./year and GHG

emissions by 512 kt $CO_2$ eq. After 2012, both GHG emissions and GHG removals showed faster-declining trends, emissions by −4.3%/year and removals by −1.2%/year. In terms of quantity, GHG removals have decreased by 504 kt $CO_2$ eq. and GHG emissions have decreased by 441 kt $CO_2$ eq.

**Table 4.** Romania's GHG emissions vs. GHG removals. Comparative synthetic indicators evolution.

| Time Segment | | 1989–2005 | 2005–2012 | 2012–2019 | 1989–2019 |
|---|---|---|---|---|---|
| GHG emissions (average), kt $CO_2$ eq./year | | 7987 | 8442 | 9086 | 8273 |
| GHG removals (average), kt $CO_2$ eq. /year | | 35,104 | 36,847 | 38,900 | 36,264 |
| GHG emissions (absolute average change), kt $CO_2$ eq./year | | 32 | 512 | −441 | 34 |
| GHG removals (absolute average change), kt $CO_2$ eq. /year | | 230 | 839 | −504 | 201 |
| GHG emissions (average annual rate), % | | 0.4 | 5.4 | −4.3 | 0.4 |
| GHG removalss (average annual rate), % | | 0.7 | 2.2 | −1.2 | 0.6 |
| GHG emissions structure on average by GHG type, % | $CO_2$ | 80.3 | 78.2 | 74.2 | 78.0 |
| | $CH_4$ | 0.03 | 0.07 | 0.08 | 0.05 |
| | $N_2O$ | 19.7 | 23.0 | 29.2 | 22.7 |

$CO_2$ emissions have a high share in the total GHG emissions in Romania's LULUCF sector. In the period 1989–2005, the share was just over 80%, 78% between 2005 and 2012, and after 2012, the share decreased to about 74% and in total, for the period 1989–2019, the $CO_2$ average was 78%. The share of $N_2O$ emissions, compared to those in the EU27 + UK, is high, representing approximately one-fifth of the total GHG emissions in Romania's LULUCF sector. Notably, there is an increasing trend for the share of $N_2O$ emissions after 2005. $CH_4$ emissions are lower than the share of this element in the EU27 + UK emissions bracket. In Romania, this share was about 0.05%, but as in the case of $N_2O$, this share practically doubled in the period 2005–2012 compared to the period 1990–2005, a trend that attenuated between 2012 and 2019 (Table 4). $CO_2$ emissions decreased from one year to another by 11 kt $CO_2$ eq., i.e., 0.2%/year. $N_2O$ emissions increased by 44 kt $CO_2$ eq., i.e., by 2.5%/year, and those of $CH_4$ increased by approximately 300 kt $CO_2$ eq., i.e., by 1.12%/year, Table 5.

**Table 5.** Comparative synthetic indicators of the evolution of GHG emissions.

| Time Segment | Emissions (Average) (kt $CO_2$ Eq./Year) | | | Absolute Average Change (kt $CO_2$ Eq./Year) | | | Average Annual Rate % | | |
|---|---|---|---|---|---|---|---|---|---|
| | $CO_2$ | $N_2O$ | $CH_4$ | $CO_2$ | $N_2O$ | $CH_4$ | $CO_2$ | $N_2O$ | $CH_4$ |
| 1989–2005 | 6412 | 1573 | 2.5 | −3 | 36 | 0.01 | −0.1 | 2.5 | 3.5 |
| 2005–2012 | 6599 | 1836 | 5.9 | 476 | 33 | 3.1 | 6.3 | 1.7 | 69.9 |
| 2012–2019 | 6746 | 2333 | 6.7 | −515 | 76 | −2.0 | −6.6 | 3.4 | −13.0 |
| 1989–2019 | 6453 | 1816 | 3.9 | −11 | 44 | 0.3 | −0.2 | 2.5 | 1.12 |

Quantitatively, $CO_2$ emissions have decreased substantially since 2012 at 515 kt $CO_2$ eq./year, exceeding the annual increase in $CO_2$ emissions recorded between 2005 and 2012. We note the increase in $N_2O$ and $CH_4$ emissions, which is worrying due to the increase in the share of these GHGs that reached after 2012 a share of 30% of total Romanian's LULUCF sector GHG emissions does not increase significantly over time. By comparison, at the EU-27 + UK level, the share of these GHGs reached a maximum of 20% in 2017 and 19% in 2019, respectively.

LULUCF GHG emissions have increased as a share of NGHGI. By 2005, the average share of GHG emissions was 4.3%; between 2005 and 2012, over about 7 years, this share

increased to 6% in the NGHGI. From 2012 to 2019, this share increased to 7.4% having an effect on the average weight at the level of the analyzed time period of 5.1%.

While GHG emissions are clearly declining at the national level, GHG emissions from the LULUCF sector show slight increase trends between 1989 and 1993, and the years 2000 and 2011. There are increases in GHG emissions in the LULUCF sector in 1994 compared to 1993 and in 2013 compared to 2011 and 2012. We also note trends of rapid decrease in 2014 compared to 2013, a trend that continues in 2015, 2016 and 2017 (Figure 7). After 2017, the trend becomes one of a relatively constant increase in GHG emissions until 2019.

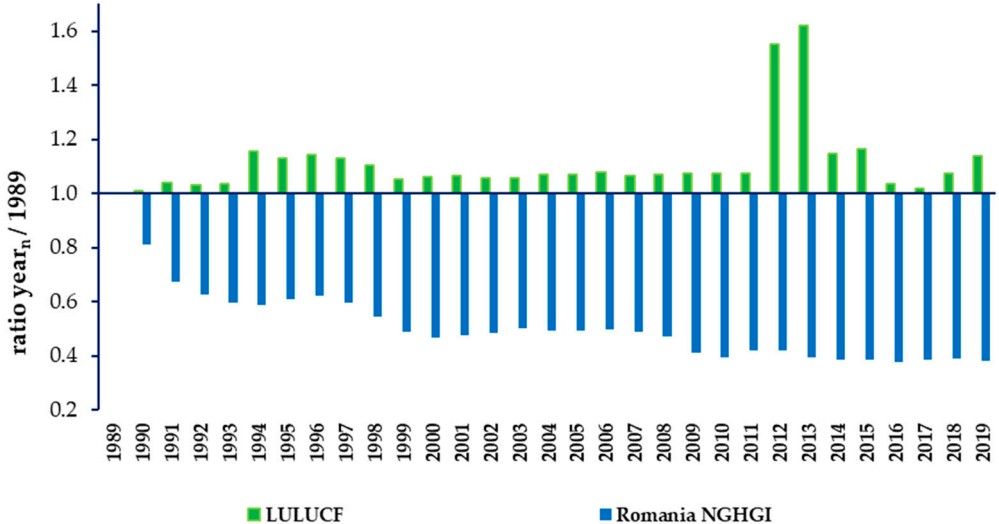

**Figure 7.** Comparative aspects of the dynamics of Romanian GHG emissions/removals between the LULUCF sector and NGHGI.

### 3.3. Romanian GHG Emissions/Removals Evolution Analysis by Land Use Categories

In Romania, the average annual increase in GHG emissions from FL Forestlands, CL—Croplands, GL—Grasslands, WL—Wetlands, SL—Settlements, OL—Other Lands, and HWP—Harvested Wood Products, a cumulative increase of approximately 58.7 kt $CO_2$ eq./year, exceeded the decrease of WL emissions, 24.8 kt $CO_2$ eq. This led to an increase in GHG emissions from the LULUCF sector from one year to the next by 0.43%/year on average from one year to another by 33.8 kt $CO_2$ eq./year. Growth is based on year-to-year increases in the land categories FL, CL, GL, WL, SL, OL, resulting in an average increase in emissions of 58.7 kt $CO_2$ eq., and the total decrease in emissions from one year to another was 25 kt $CO_2$ eq. (WL), Figure 8. These land use categories contributed to the increase of emissions from the LULUCF sector by 52.7% with an average weight for the analyzed time horizon of approximately 15% in the total GHG emissions from the LULUCF sector.

The contribution of these land use categories to the annual average increase in GHG emissions was about 47%. Until 2012, the share of emissions from the SL was on average about 48%, and after this, the share increased to 52.4%, representing half of the total GHG emissions of the sector. This fact determined that in the 1989–2019 time period, the average weight should be approximately 48.3%. OL GHG emissions had an average share of between 20% and 21.5% in the period 1989–2012, and after this period, the share decreased to 17.6% of the total GHG emissions of the sector. Regarding WL emissions, they had an average share of 18.3% of the total GHG emissions at the level of LULUCF in the period 1998–2005, after 2005 this share decreased steadily to 15.2% in the period 2005–2012, and after 2012 this share decreased to 6.5%. GL emissions increased as a share from 5.6% in 1989–2005 to 9.7% between 2005 and 2012, and after 2012 this share of the trend increased to an average of about 14.4%.

GHG emissions from CL recorded an average share in total GHG emissions for the analyzed time period of 1.3%. The emissions from FL have a weight of less than 0.11%,

share characteristics of the period 2012–2019, and a double weight compared to the period 1989–2005.

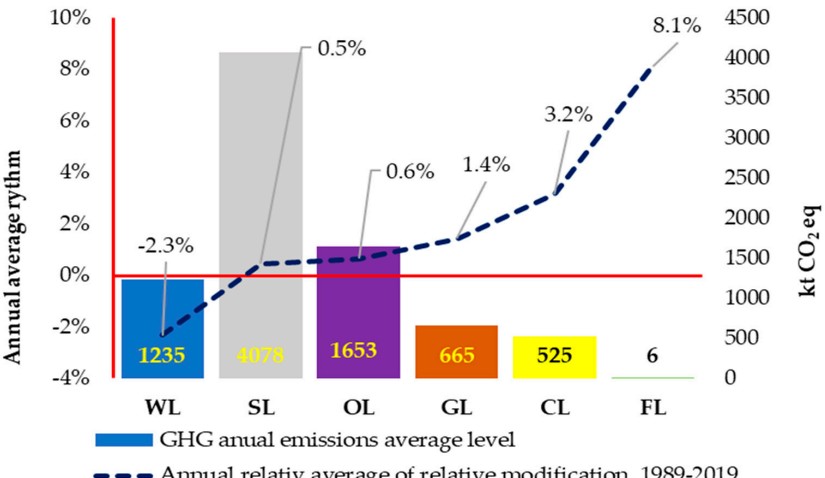

**Figure 8.** Romania—GHG emissions/removals dynamics comparative aspects, by land use category. Note 3: FL—Forestlands; CL—Croplands; GL—Grasslands; WL—Wetlands; SL—Settlements; OL—Other Lands; HWP—Harvested Wood Products.

In 2019, the volume of GHG removals generated by the LULUCF sector was 38,700 kt CO$_2$ eq., which was approximately 6000 kt CO$_2$ eq., 18.4% higher than that recorded in 1989, approximately 32,700 kt CO$_2$ eq. Specific activities to the subcategories of land use CL and HWP in 2019 generated an increase in GHG removals compared to 1989, with just over 9000 kt of CO$_2$ eq., while FL caused a decrease in GHG removals with about 3000 kt of CO$_2$ eq.

Evolution in GHG removals in the LULUCF sector was relatively constant during the period 1989–2011, as it is directly influenced by the variables used, namely: (i) activity data/AD (kha) obtained by mathematical modeling, linear interpolation and extrapolation; (ii) emission factors/EF characteristic of carbon pools, living biomass/LB, dead organic matter/DOM and Soil, type Tier 1 (IPCC 2006); (iii) classification in the Boreal climate zone (IPCC 2006) [13]. The GHG removals increased rapidly in the 2012–2013 time period, generating a spike, after which there was a decreasing trend until 2017, and after 2017 there was a decreasing trend until 2019 (Figure 9). GHG removal for the 2012–2019 time period stems from the following drivers: (i) activity data/AD (kha) estimated following the approach 3/geospatially investigations, LPIS/IACS + CLC + LiDAR and aero-photogrammetry technologies; (ii) emission factors/EF characteristic of carbon pools, country specific/CS and default/D (IPCC 2006 and IPCC Refinement 2019), respectively Tier 1 and Tier 2; (iii) classification in the Warm temperate climate zone (IPCC 2006) [13].

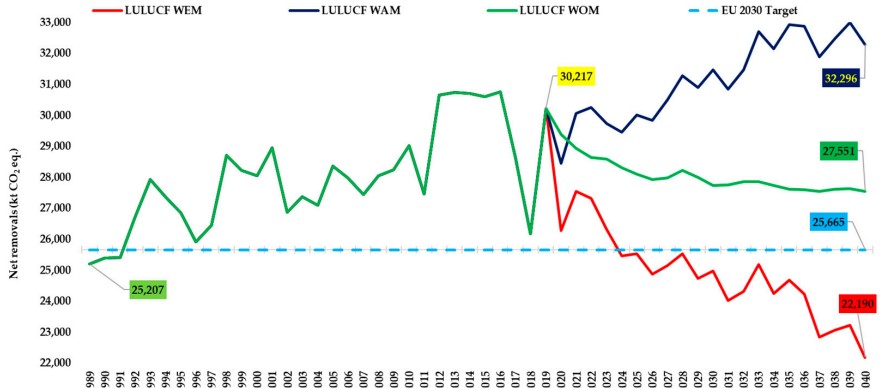

**Figure 9.** LULUCF GHG emissions/removals scenarios vs. EU 2030 target.

### 3.4. Romanian LULUCF Sector GHG Emissions/Removals Projections by 2040

The projections of quantitative levels were based on linear models in which area and time variables were assimilated as factorial variables. The Data Analysis was used to determine the parameters of the linear models, test the significance of the regression coefficients, and validate the regression model. In order to estimate the GHG emissions/removals projections for the 2020 to 2040 time period, three scenarios were generated: the reference scenario (WEM), the scenario with measures (WAM) and the scenario with additional measures (WOM).

LULUCF-specific activities may lead to GHG emissions or GHG removals and carbon stock changes/CSC in related land use categories. For the 2021–2030 time period, the European Commission aims to enshrine the so-called "no-debit rule" in EU law, by incorporating LULUCF for the first time into the EU's efforts to reduce emissions [12]. The results of the projections were obtained from the processing of existing information from the NGHGI 2021 according to the methodology presented above. After analyzing all the information, the following scenarios resulted: (i) the WEM scenario is based on the historical evolution of the GHG emissions/removals in the LULUCF sector for the 1989–2019 time period for each predefined land use category, using historical data from the NGHGI; (ii) the WAM scenario is based on 2005, the year in which the Kyoto Protocol was implemented. The projections were based on the policies and measures implemented or started in the 2005–2019 time period; (iii) the WOM scenario is based on the WAM scenario, to which were added the annual average level and the annual average change, calculated on the basis of the annual values of the 2012–2019 time period (Table 6) [26,27].

**Table 6.** Romanian's LULUCF sector GHG emissions/removals evolution, by scenario.

| Scenarios Year | WEM Scenario (kt $CO_2$ Eq.) | WAM Scenario (kt $CO_2$ Eq.) | WOM Scenario (kt $CO_2$ Eq.) |
|---|---|---|---|
| 1989 | −25,207 | −25,207 | −25,207 |
| 2005 | −28,370 | −28,370 | −28,370 |
| 2012 | −30,658 | −30,658 | −30,658 |
| 2019 | −30,217 | −30,217 | −30,217 |
| 2020 | −26,288 | −28,458 | −29,393 |
| 2025 | −25,541 | −30,021 | −28,105 |
| 2030 | −24,989 | −31,468 | −27,749 |
| 2035 | −24,691 | −32,939 | −27,618 |
| 2040 | −22,190 | −32,296 | −27,551 |

Following the generation of the three types of projections, a particular dynamic is observed (Table 6): (i) for the WEM scenario, the net GHG removals would decrease by approximately 218 kt $CO_2$ eq., i.e., by 0.9%, in 2030 compared to the reference year, and in 2040 compared to 1989, the decreasing trend would be accentuated both in terms of absolute values, a decrease of more than 3000 kt $CO_2$ eq., as well as in relative values of 12%; (ii) in the case of the WAM scenario with ~6300 kt $CO_2$ eq. in 2030 compared to 1989, the trend would be to increase by 2040 by just over ~7000 kt $CO_2$ eq. compared to the reference year; (iii) in the case of the WOM scenario, the increases are about one-third of those estimated in the WAM scenario, with an increase of 2500 kt $CO_2$ eq. in 2030 compared to 1989, and in 2040 the increase compared to the reference year would be about 2300 kt $CO_2$ eq.

According to Figure 9, the values of the projections under the WAM and WOM scenarios are in the national target of net GHG removals, proposed to be achieved in 2030, in accordance with the LULUCF Regulation Proposal of 25,665 kt $CO_2$ (in 2030), assuming that the measures and policies implemented are the right drivers.

Using the methodology presented above, the projected data on the GHG emissions and GHG removals are obtained (Table S2, Supplementary Materials). The LULUCF sector has higher GHG removals values compared to GHG emissions and in terms of dynamics, GHG removals have a lower upward trend than the upward trend of GHG emissions in all three scenarios developed (Figure 10).

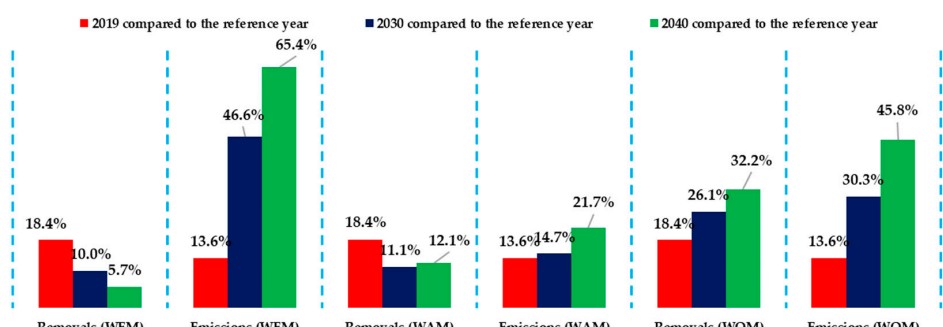

**Figure 10.** Comparative dynamics of the forecasted evolution of LULUCF GHG emissions vs. GHG removals scenarios.

## 4. Conclusions

The study presents the evolution of LULUCF sector GHG emissions/removals in EU-27 + UK for the 1990–2019 time period, as well as projections for Romania, until 2040. The results indicated that in the EU-27 + UK, the LULUCF sector net removals increased from around 193,000 kt $CO_2$ eq. to 328,900 kt $CO_2$ eq. in 1999, reaching over 243,000 kt $CO_2$ eq. In the 1990–2019 time period, the average removals were approximately 289,800 kt $CO_2$ eq./year, a volume that increased from one year to another to 30,400 kt $CO_2$ eq. In relative terms, during the 1990–2019 time period, the GHG removals increased by 0.8%/year.

The LULUCF sector differs from other sectors in that it can act as both a source and a sink. In Romania, the LULUCF sector net GHG removals had increased since 1989, from 25,200 kt $CO_2$ eq. to 30,200 kt $CO_2$ eq. in 2019, respectively, an increase of approximately 20%. Furthermore, for the LULUCF sector, the annual GHG emissions/removals average was approximately 28,000 kt $CO_2$ eq., representing approximately 9.7% of the annual average at the EU's level (around 290,000 kt $CO_2$ eq. calculated for the period 1990–2019). Romania, along with 22 other countries, has a net sink behavior. GHG projections were performed based on linear models in which area and time were assimilated as factor variables. In the case of the WAM scenario with 6300 kt $CO_2$ eq. in 2030 compared to 1989; the projection is expected to increase by 2040 by just over 7000 kt $CO_2$ eq. compared to the reference year. In the WOM scenario, the increases are about one third of those estimated in the WAM scenario, with an increase of 2500 kt $CO_2$ eq. in 2030 compared to 1989, and in 2040 the increase compared to the reference year would be around 2300 kt $CO_2$ eq. In terms of dynamics, GHG removals have a lower upward trend than the upward trend of GHG emissions in all three scenarios developed. In comparison, GHG emissions from the LULUCF sector show high trends in terms of projected dynamics after 2019.

The current photography of the GHG emissions/removal levels in the Romanian LULUCF sector shows that the application of the measures and policies carried out in the 2005–2019 time period will achieve the target established by the LULUCF Regulation proposal by 2030–2040. The study, based on current data from the Romanian National GHG Inventory, developed three different types of projections, WOM, WEM, WAM, until 2040, following the climate neutrality targets set by the European Commission in an attempt to validate or not policies and measures taken by National Authorities. We believe that this type of matrix analysis can be a useful tool, is easy to multiply/replicate for other countries, and is part of the reporting under the Annex I/UNFCCC (United Nations Framework

Convention on Climate Change) in their tendency to assess emission/removal levels in relation to their own GHG benchmarks until 2040.

**Supplementary Materials:** The following supporting information can be downloaded at: https://www.mdpi.com/article/10.3390/atmos13101638/s1, Figure S1: The net removals dynamics, EU-27 + UK (kt CO2 eq.); Figure S2: The GHG emissions structure evolution, by GHG type, EU-27 + UK; Figure S3: GHG emissions *vs.* GHG removals average annual rate, by GHG type, EU-27 + UK; Figure S4: GHG emissions vs. GHG removals average annual rate, by GHG type, EU-27 + UK; Table S1: The LULUCF sector GHG emissions evolution, EU-27 + UK (kt $CO_2$ eq.); Table S2: Romanian's LULUCF sector GHG emissions vs. GHG removals evolution, by scenario (kt $CO_2$ eq.).

**Author Contributions:** M.I., M.C. wrote the original draft, and all authors contributed to the analysis; writing—review and editing, R.E.I.; figures were created by R.E.I.; R.G. coordinated the analysis for this paper; the policy inventory and database were created by M.C. and F.B.; scenario database was coordinated by M.I., R.G., R.Z., A.C., A.Z. and A.M.I.; resources, R.E.I.; visualization, O.R.B.; supervision, M.C. All authors have read and agreed to the published version of the manuscript.

**Funding:** This research was funded by the Ministry of Research, Innovation and Digitization through Contract no. 9N/2019, Project PN 19 11 03 01/2019–2022: "Studies on the obtaining and improvement of the acido-basic properties of the nanoporous catalytic materials for application in wastes valorisation" and Program 1-Development of the national research and development system, Subprogram 1.1. Institutional performance-Projects to finance excellence in RDI, Contract No. 19PFE/30.12.2021 and a grant of the Environmental Fund Administration.

**Institutional Review Board Statement:** Not applicable.

**Informed Consent Statement:** Not applicable.

**Data Availability Statement:** The data presented in this study are available on request from the corresponding authors. The data are not publicly available due to institutional policies.

**Acknowledgments:** Contract 49/N/19 September 2019—additional act 1/4 May 2020 in accordance with GD no. 590/2019 for the definition of the obligations of administration of the subdomain Land Use, land use change and forestry (LULUCF), part of the Climate Change domain.

**Conflicts of Interest:** The authors declare no conflict of interest.

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
