# Peer review of "The GHGs Evolution of LULUCF Sector at the European Union (EU-27 + UK): Romania Case Study"

_atmosphere, doi:10.3390/atmos13101638_

Round 1
Reviewer 1 Report
The GHGs evolution of LULUCF sector at the European Union (EU-27 + UK):
Romania case study Iordache et al.,
General Comments
The paper reports on the evolution of Land Use, Land Use Change, and Forestry (LULUCF) greenhouse gas (GHG) emission removals at the EU-27 +UK level for the 1990-2019 time period, as well as LULUCF emissions/removals forecasts for Romania up to 2040. Following the first round of reviews, the authors have made several changes which have improved the paper as compared to the initial submission. However, after another thorough and careful look at the results and discussion section, I am beginning to wonder if this work is really suitable for consideration in the Atmosphere journal. For example, most of the data presented (Table 1, Figure 1, Figure 4, etc) cannot really be regarded as original research data. Such material is available publicly from the inventories. I do not want to be mean or overly critical about this work, but sincerely speaking, this manuscript is more of a report than a manuscript. There is no robust scientific discussion of the results. Throughout the result and discussion, I do not see any novel scientific or atmospheric chemistry discussion that is suitable for publication in a high impact journal like Atmosphere. Even if the paper is to be viewed like a report (say EPA report for example) the authors did not do a thorough job in terms of presenting Figures that can give a coherent explanations. The Figures (and their presentation style) are not the typical Figures and Tables that can be presented in a research journal.
I encourage the authors to look at these comments carefully. After looking at my comments, I encourage the authors to sample a few research papers from the Atmosphere journal. This way they can compare the quality of manuscripts accepted in the Atmosphere journal with their work. I think this is the best way to go about it without the authors feeling like the paper was harshly reviewed.
Author Response
Dear reviewer,
We would like to thank you for a careful and thorough reading of the manuscript “The GHGs evolution of LULUCF sector at the European Union (EU-27 + UK): Romania case study” by Iordache et. al., and for the thoughtful comments and constructive suggestions, which help to improve the quality of this draft. According to your recommendations, we have revised our manuscript, and its final version is enclosed. Point-by-point responses to the comments are listed below.
Q1. The paper reports on the evolution of Land Use, Land Use Change, and Forestry (LULUCF) greenhouse gas (GHG) emission removals at the EU-27 +UK level for the 1990-2019 time period, as well as LULUCF emissions/removals forecasts for Romania up to 2040. Following the first round of reviews, the authors have made several changes which have improved the paper as compared to the initial submission. However, after another thorough and careful look at the results and discussion section, I am beginning to wonder if this work is really suitable for consideration in the Atmosphere journal. For example, most of the data presented (Table 1, Figure 1, Figure 4, etc) cannot really be regarded as original research data. Such material is available publicly from the inventories. I do not want to be mean or overly critical about this work, but sincerely speaking, this manuscript is more of a report than a manuscript. There is no robust scientific discussion of the results. Throughout the result and discussion, I do not see any novel scientific or atmospheric chemistry discussion that is suitable for publication in a high impact journal like Atmosphere. Even if the paper is to be viewed like a report (say EPA report for example) the authors did not do a thorough job in terms of presenting Figures that can give a coherent explanation. The Figures (and their presentation style) are not the typical Figures and Tables that can be presented in a research journal.
R1. In fact, the data used and processed in this work, presented in Tables 1-3 and Figures 1-6, respectively, are taken from EUROSTAT (see reference no. 24), in order to be able to observe the contribution of EU 27 + UK in terms of the amount of GHG emissions/removals, in the 1990-2019 time period. Starting with table 4 to table 7, respectively Figure 7 to Figure 10, the data used come from the National Inventory of Greenhouse Gas Emissions (NGHGI Romania 2021). We believe that this work helps the scientific society, and not only, to get an idea of ​​the situation of GHG’s evolution at the level of the EU and Romania, respectively. Concretely, we are offering a calculation methodology, the absolute mean change method and the mean index method, through which the Party's can realize their projections (WOM, WAM, WEM) of GHG emissions/removals until 2040, in order to find optimal solutions to mitigate the climate change impact through specific measures and policies. There are not many scientific works with this specific, that's why I considered this approach beneficial. Also, we have updated a series of Figures from the article text, at your suggestion, respectively Fig 1, Fig 2, Fig 3, Fig 4, Fig 5, Fig 6, Fig 7, Fig 8, Fig 9, Fig 10, for a better presentation and understanding of the processed data.
Q2. I encourage the authors to look at these comments carefully. After looking at my comments, I encourage the authors to sample a few research papers from the Atmosphere journal. This way they can compare the quality of manuscripts accepted in the Atmosphere journal with their work. I think this is the best way to go about it without the authors feeling like the paper was harshly reviewed.
R2. We have looked at MDPI Atmosphere for some articles from our field of work and they helped us to adjust our initial figures and tables, in accordance with your recommendations.
Thank you very much for your valuable comments that have improved our paper.
On behalf of the authors,
Marius CONSTANTINESCU

Reviewer 2 Report
I checked manuscript and response provided. The authors have corrected manuscript and revised version satisfy my requirements. Authors have implemented my comments and provided relevant answers. I do not have anything against for publishing paper in current version.
Author Response
Dear reviewer,
We would like to thank you for a careful and thorough reading of the manuscript “The GHGs evolution of LULUCF sector at the European Union (EU-27 + UK): Romania case study” by Iordache et. al., and for the thoughtful comments and constructive suggestions, which help to improve the quality of this draft. According to your recommendations, we have revised our manuscript, and its final version is enclosed. Point-by-point responses to the comments are listed below.
Q1. I checked manuscript and response provided. The authors have corrected manuscript and revised version satisfy my requirements. Authors have implemented my comments and provided relevant answers. I do not have anything against for publishing paper in current version.
R1. Thank you for your valuable comments that have improved our paper.
On behalf of the authors,
Marius CONSTANTINESCU

Reviewer 3 Report
The manuscript is useful, actual and has interest for experts and researchers in the field of emissions and removals dynamics.
I would recomend:
- to add a general list of abbreviations (i.e. EST at the page 2),
- to check picture caption for Figure 2. GHG emissions vs. GHG removals dynamics, by GHG type, EU-27 + UK (Emissions de CH4) - page 9.
- for page 14, last paragraph:
"We have increases in GHG emissions in the LUL14UCF sector in 1994 compared to 1993 and in 2013 compared to 2011 and 2012" - probably could "there are increases..." could be better (page 14)
- for Figure 8 – to give once more explanation of abbreviations
Author Response
Dear editors and anonymous reviewers,
We would like to thank you for a careful and thorough reading of the manuscript “The GHGs evolution of LULUCF sector at the European Union (EU-27 + UK): Romania case study” by Iordache et. al., and for the thoughtful comments and constructive suggestions, which help to improve the quality of this draft. According to your recommendations, we have revised our manuscript, and its final version is enclosed. Point-by-point responses to the comments are listed below.
Q1. I would recomend: to add a general list of abbreviations (i.e. EST at the page 2),
R1. We have added in the text what "ETS" and "ESR" mean.
Q2. Check picture caption for Figure 2. GHG emissions vs. GHG removals dynamics, by GHG type, EU-27 + UK (Emissions de CH4) - page 9.
R2. We replaced the word "de" in Figure 2.
Q3. For page 14, last paragraph: "We have increases in GHG emissions in the LUL14UCF sector in 1994 compared to 1993 and in 2013 compared to 2011 and 2012" - probably could "there are increases..." could be better (page 14)
R3. We replaced the "We have increases” with “There are increases”.
Q4. For Figure 8 – to give once more explanation of abbreviations
R4. We added "Note 3" in which we explained the abbreviations in Figure 8.
Thank you very much for your valuable comments that have improved our paper.
On behalf of the authors,
Marius CONSTANTINESCU

Round 2
Reviewer 1 Report
I still stand by my earlier comments that I gave the authors a few days ago. I do not think this manuscript is suitable for publication in a scientific or Atmospheric Journal like Atmosphere. This is more like a report than a scientific piece. As an atmospheric scientist in the field of fundamental and applied atmospheric chemistry, I do not see how such a report can be considered as a scientific paper. I have reviewed more than fourty manuscripts in the field of Physical Chemistry and Atmospheric Chemistry and I really can’t recommend this paper for publication in Atmosphere. It was initially submitted in the International Journal of Environmental Research and Public Health (JERPH) and rejected. If it can be rejected in such a public health journal that has a lesser Atmospheric significance that the Atmospheric Journal, that tells you the manuscript has major flaws.
This is my genuine, solid scientific view and I strongly stand by it. If the authors want to know the what content is suitable for publication in Atmosphere journal, they should sample a few Atmosphere and compare with their work. If they are genuine in their submission, they will be able to see that their work is far from the scope of the Atmospheric journal. Even for a report, the Figures in this paper are unacceptable, the way they are presented.